# HID-1 is required for homotypic fusion of immature secretory granules during maturation

Wen Du[1†], Maoge Zhou[1,2†], Wei Zhao[1,2†], Dongwan Cheng[1†], Lifen Wang[1], Jingze Lu[1], Eli Song[1], Wei Feng[1], Yanhong Xue[1*], Pingyong Xu[2,3*], Tao Xu[1,2*]

[1]National Laboratory of Biomacromolecules, CAS Center for Excellence in Biomacromolecules, Institute of Biophysics, Chinese Academy of Sciences, Beijing, China; [2]College of Life Sciences, University of Chinese Academy of Sciences, Beijing, China; [3]Key Laboratory of RNA Biology, Institute of Biophysics, Chinese Academy of Sciences, Beijing, China

**Abstract** Secretory granules, also known as dense core vesicles, are generated at the trans-Golgi network and undergo several maturation steps, including homotypic fusion of immature secretory granules (ISGs) and processing of prohormones to yield active peptides. The molecular mechanisms governing secretory granule maturation are largely unknown. Here, we investigate a highly conserved protein named HID-1 in a mouse model. A conditional knockout of HID-1 in pancreatic β cells leads to glucose intolerance and a remarkable increase in the serum proinsulin/insulin ratio caused by defective proinsulin processing. Large volume three-dimensional electron microscopy and immunofluorescence imaging reveal that ISGs are much more abundant in the absence of HID-1. We further demonstrate that HID-1 deficiency prevented secretory granule maturation by blocking homotypic fusion of immature secretory granules. Our data identify a novel player during the early maturation of immature secretory granules.

**\*For correspondence:**
xueyanhong@moon.ibp.ac.cn (YX); pyxu@ibp.ac.cn (PX); xutao@ibp.ac.cn (TX)

[†]These authors contributed equally to this work

**Competing interests:** The authors declare that no competing interests exist.

## Introduction

The trans-Golgi network (TGN) acts as the central transport and sorting station within cells, and is responsible for proper segregation and sorting of newly synthesized membrane and soluble cargo proteins to their appropriate subcellular destinations. Secretory granules (SGs, also known as dense core vesicles, DCVs) are specialized organelles of the regulated secretory pathway and play crucial roles in a wide variety of biological processes, including organism development, synaptic plasticity, immunity and metabolism. After passing through the TGN, peptide hormones and neuropeptides are first packaged and stored in immature secretory granules (ISGs). ISGs then undergo multiple maturation steps to become mature secretory granules (MSGs). These maturation steps include 'sorting-by-entry' of SG-specific cargoes, homotypic fusion of ISGs, acidification of granular lumen, condensation and processing of peptide hormones by prohormone convertases PC1/3 and PC2, and 'sorting-by-retention' (or 'sorting-by-exit') to remove cargoes not destined for SGs (*Arvan and Castle, 1998*). Homotypic fusion of ISGs has been observed in several neuroendocrine cells, including PC12 cells (*Ahras et al., 2006*; *Tooze et al., 2001*; *Wendler et al., 2001*), pituitary cells (*Farquhar and Palade, 1981*) and mast cells (*Hammel and Meilijson, 2015*), and in the formation of large dense core vesicles in yeast (*Asensio et al., 2013*), but it has not been observed in pancreatic β cells (*Arvan and Halban, 2004*). Hence, it remains unclear whether homotypic fusion is universally required for SG maturation.

Besides the significance of homotypic fusion of ISGs, the molecular mechanisms underlying ISG fusion and maturation remain elusive (*Kogel and Gerdes, 2010*). SNARE proteins (soluble N-ethyl-maleimide-sensitive fusion protein (NSF) attachment protein (SNAP) receptors) are essential components of the highly conserved machinery involved in membrane fusion (*Jahn and Scheller, 2006*). An in vitro fusion assay has revealed that ISG homotypic fusion is dependent on NSF and α-SNAP (*Urbe et al., 1998*) and on the SNARE protein syntaxin 6 (STX6), but not on syntaxin 1 or SNAP-25 (*Wendler et al., 2001*). It has been suggested that Syt-IV, a synaptotagmin protein that interacts with SNAREs, localizes to ISGs and is involved in SG maturation (*Ahras et al., 2006*). While SNAREs are essential components of membrane fusion, they often require other accessory proteins to facilitate the assembly of SNARE complexes and to coordinate the tethering/docking and priming of vesicles (*Cai et al., 2007*; *Pfeffer, 1999*). The molecular players involved in tethering/docking and priming of ISGs are not known.

A gene designated as *hid-1* was identified during a search for mutants with a high-temperature-induced dauer formation (Hid) phenotype in *Caenorhabditis elegans* (*Ailion and Thomas, 2003*). Subsequently, *hid-1* has been isolated by a genetic screen for mutants mislocalizing rabphilin (RBF-1 in *C. elegans*), an effector of RAB-27 that is involved in SG exocytosis (*Mesa et al., 2011*). The *Hid1* gene encodes a highly conserved protein (HID-1) that has homologs in *Drosophila melanogaster*, mouse, and *Homo sapiens*, but not in budding yeast. HID-1 shares no homology domains with other proteins but is suggested to be a membrane-attached protein that shuttles between membrane and cytosol (*Wang et al., 2011*). Recent studies in *C. elegans* have suggested an essential role for HID-1 in neuropeptide signaling (*Mesa et al., 2011*; *Yu et al., 2011*), but the exact site(s) of action and the molecular mechanism by which HID-1 acts remain unknown.

In this study, we generated a conditional knockout (KO) mouse model in which the mouse *Hid1* gene is disrupted specifically in pancreatic β cells. This model allowed us to define a previously unidentified cellular function of the HID-1 protein. We demonstrate that HID-1 is a novel factor required for homotypic fusion of ISGs. Loss of function of HID-1 in mice leads to diabetes-like symptoms characterized by glucose intolerance, insufficient insulin release, and elevated proinsulin secretion.

## Results

### Knockout of HID-1 in pancreatic β cells causes glucose intolerance

To avoid embryonic lethality and complications in interpreting the data from global KO mice, we employed a conditional gene-targeting approach to derive mice that selectively lack HID-1 expression in pancreatic β cells. We first generated *Hid1*-floxed mice by introducing two LoxP sites into the intronic region flanking exons 2–6 of the mouse *Hid1* gene (*Figure 1A*). Then, *Hid1*-floxed mice were mated with mice overexpressing Cre recombinase under the control of the rat insulin 2 gene promoter (RIP-Cre mice) to produce cell-specific HID-1 KO mice (Hid1-betaKO, homozygous for *Hid1^{flox/flox}* and heterozygous for RIP-Cre). As demonstrated in *Figure 1B*, HID-1 protein was greatly reduced in islets isolated from Hid1-betaKO mice compared to that in age-matched RIP-Cre mice, which were used thereafter as wild-type (WT) controls. The remaining expression of HID-1 in KO islets was only 20% of that from WT islets (*Figure 1C*). By positive immunofluorescence staining, we verified that HID-1 was highly expressed in pancreatic β cells, but only weakly expressed in α cells (*Figure 1D*, *Figure 1—figure supplement 1*). Histogram analysis of HID-1 levels in individual β cells confirmed the largely reduced expression of HID-1 in KO β cells (*Figure 1—figure supplement 1B*). Hid1-betaKO mice are viable and fertile, and the mice showed no obvious phenotype with respect to body weight (*Figure 1E*), feeding, mating performance and overall behavior. To determine whether the disruption of HID-1 in β cells corresponds to abnormalities in the temporal control of glucose metabolism, we analyzed glucose profiles upon intraperitoneal (i.p.) administration of glucose or insulin. At 10–12 weeks of age, Hid1-betaKO mice exhibited higher basal glucose levels and glucose intolerance than RIP-Cre mice during glucose-tolerance testing (GTT, *Figure 1F*). However, Hid1-betaKO mice performed normally in insulin-tolerance testing (ITT) (*Figure 1G*). These results indicate that impaired glucose tolerance may be caused by insufficient insulin release.

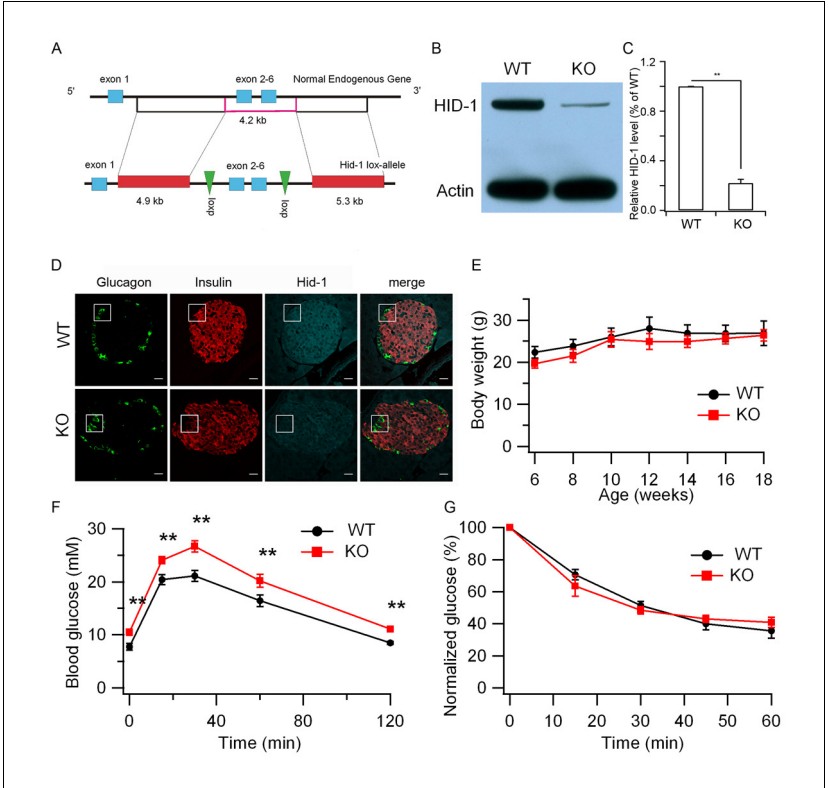

**Figure 1.** Hid1-betaKO mice exhibit glucose intolerance. (**A**) A schematic diagram showing the endogenous mouse *Hid-1* gene and the targeting floxed construct. (**B**) Western blot analysis of mouse HID-1 and actin in isolated islets from WT and Hid1-betaKO (KO) mice (five biological replicates from ten mice). (**C**) Quantitative analysis of the reduction of HID-1 protein in KO islets. (**D**) Immunofluorescent histochemical analysis of pancreas sections of WT and KO mice, showing glucagon (green), insulin (red) and HID-1 (cyan). The white squares were enlarged in *Figure 1—figure supplement 1A* for a magnified view. Scale bar: 20 μm. (**E**) Comparison of body weight (n = 6, three males and three females in each group) and (**F**) blood glucose during GTT (i.p. administration of 2 g/kg body weight glucose. WT, n = 8, four males and four females; KO, n = 11, five males and six females. p=0.003, 0.005, 0.001, 0.02 and 0.002 at 0, 15, 30, 60 and 120 min, respectively, t-test.) and (**G**) normalized blood glucose during ITT (i.p. administration of 0.5 U/kg body weight insulin. WT, n = 9, four males and five females; KO, n = 10, five males and five females) between WT and KO mice.

The following figure supplements are available for figure 1:

**Figure supplement 1.** Quantification of HID-1 expression by immunofluorescence in pancreas sections.

**Figure supplement 2.** Hid1-betaKOmice show normal islet and β-cell morphology.

**Figure supplement 3.** HID-1 deficiency does not impair depolarization-induced SG exocytosis.

## Hid1-betaKO mice show normal islet and β cell morphology

Insufficient insulin release may arise from abnormalities in β cell development, SG biogenesis, insulin production, or exocytosis. To distinguish these possibilities, we first examined the morphology of the pancreatic islets, which was comparable between age-matched WT and KO islets (*Figure 1—figure supplement 2A*). Quantification indicated that the islet size (*Figure 1—figure supplement 2C*) and the density of the islets (*Figure 1—figure supplement 1D*) in KOmice were not significantly different from those of in WT mice. Furthermore, the general morphology of β cells identified with pro-insulin staining (*Figure 1—figure supplement 2B*) and the density of β cells (*Figure 1—figure supplement 2E*) were similar in the WT and KO mice.

## Increased proinsulin secretion in Hid1-betaKO mice

To address whether HID-1 is required for SG exocytosis, we first employed membrane capacitance ($C_m$) measurement, which primarily measures the ability of SGs to fuse with the plasma membrane (PM) of β cells (*Kanno et al., 2004*). As shown in *Figure 1—figure supplement 3*, no significant difference was observed between WT and KO β cells for either $Ca^{2+}$ influx or the $Ca^{2+}$-induced $C_m$ increase. Consistent with the fact that HID-1 is not required for SG exocytosis, *hid-1* mutants did not block secretion of a DCV cargo (AEX-5-VENUS) in *C. elegans* (*Mesa et al., 2011*). Collectively, these results suggest that HID-1 is not required for the exocytosis of SGs per se in either mouse or worm.

Next, we directly measured insulin release from β cells. Interestingly, the serum proinsulin level was markedly elevated from 5 pM in WT mice to 78 pM in Hid1-betaKOmice (*Figure 2A*), whereas the basal insulin level was only slightly increased (150 ± 29 pM for WT and 198 ± 39 pM for KO mice). The ratio between proinsulin and insulin was determined to be 3.3%, consistent with previous reports (*Wijesekara et al., 2010*; *Zhu et al., 2002*). This value increased to 39.4% in KO mice (*Figure 2B*). The increase of proinsulin may be partially caused by the prolonged plasma half-life of proinsulin, as compared with insulin, in the circulation. To address this possibility, we directly measured glucose-stimulated insulin secretion (GSIS) from isolated islets. We found that isolated size-matched islets from Hid1-betaKO mice released much less insulin (*Figure 2C*), but more proinsulin (*Figure 2D*), in response to glucose stimulation. It has been reported that RIP-Cre mice have Cre expression in a population of hypothalamic neurons (*Lee et al., 2006*). To rule out the possibility that the elevated proinsulin phenotype in Hid1-betaKO mice was secondary to HID-1 function in

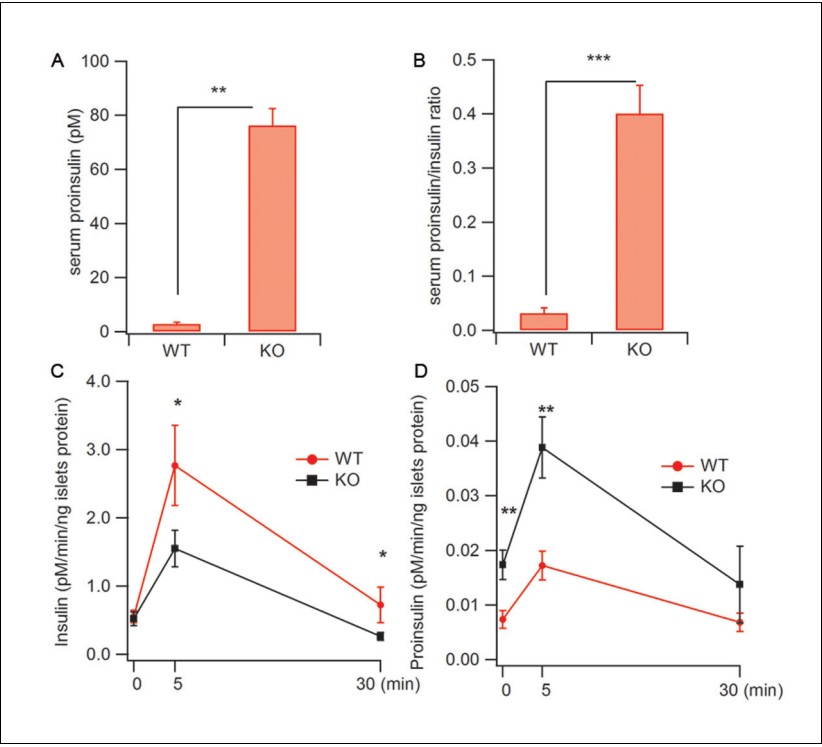

**Figure 2.** Increased proinsulin secretion in Hid1-betaKO mice. (**A**) Hid1-betaKO (KO) mice exhibit a high basal serum proinsulin level and (**B**) a high proinsulin/insulin ratio under normal feeding conditions (WT, n = 9, five males and four females; KO, n = 17, nine males and eight females; p = 0.0011 and 0.0005 for A and B, respectively, t-test). (**C**) Glucose (20 mM) stimulates insulin and (**D**) proinsulin secretion from isolated islets of WT and KO mice (four biological replicates from eight mice, p = 0.022 and 0.02 for 5 and 30 min, respectively, in (**C**); and p = 0.004 and 0.005 for 0 and 5 min. in (**D**), respectively; t-test).

The following figure supplement is available for figure 2:

**Figure supplement 1.** Pan-neuronal ablation of HID-1 fails to increase serum proinsulin.

neurons, we generated HID-1 KO in all neurons employing Nestin-Cre mice. Pan-neuronal ablation of HID-1 failed to increase proinsulin in the serum (*Figure 2—figure supplement 1*), suggesting that the elevated proinsulin phenotype was mainly due to lack of HID-1 function in pancreatic β cells.

## HID-1 deficiency causes a defect in proinsulin processing

The reduction of insulin release could be due to defects in proinsulin processing or to a preferential fusion of ISGs. These scenarios are predicted to result in lower levels of insulin or in the accumulation of insulin, respectively. To distinguish between these possibilities, we verified the levels of proinsulin and mature insulin within β cells. Western blot analysis demonstrated that islets from Hid1-betaKO mice contained a lower level of mature insulin but an elevated level of proinsulin compared with WT islets (*Figure 3A*). We then characterized the kinetics of proinsulin processing to insulin. Islets from WT or KO mice were labeled with [$^{35}$S]-cysteine in high glucose for 1 hr and then chased in low glucose for a total of 3 hr. We observed a higher amount of proinsulin and delayed processing of proinsulin in Hid1-betaKO mice (*Figure 3B,C*). As a consequence, the insulin/proinsulin ratio was significantly lower in KO islets than in WT controls during the chase period (*Figure 3D*).

Insulin consists of two peptide chains, A and B, linked by disulfide bonds. Conversion of proinsulin to insulin involves cleavage at two sites to remove the C-peptide that links the B and A chains

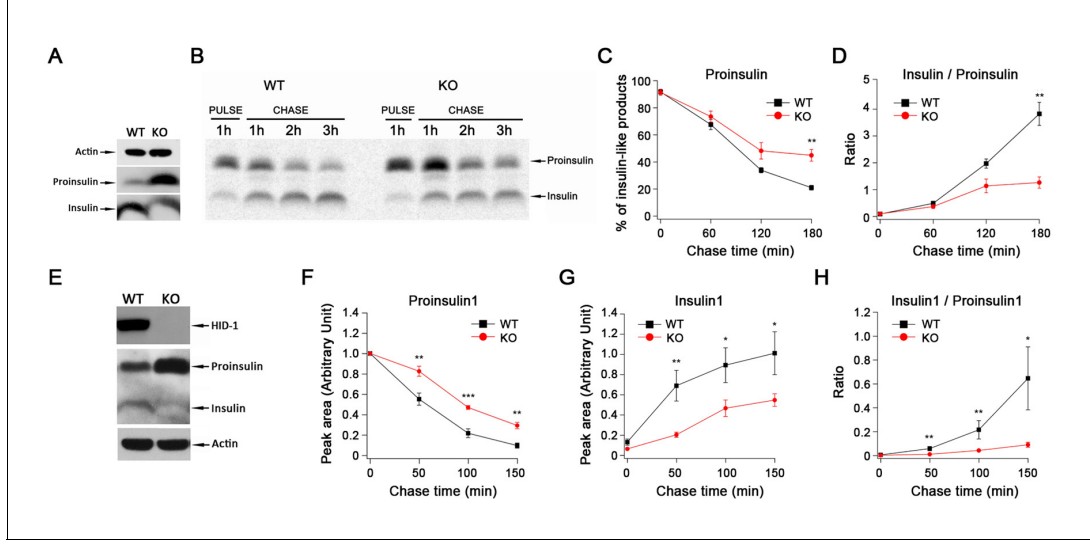

**Figure 3.** HID-1 deficiency causes a defect in proinsulin processing. (**A**) Western blotting of islets from WT and KO mice with an anti-insulin polyclonal antibody. The proinsulin level in KO mice is higher than that in WT mice. Results are representative of three experiments from six mice. (**B**) Pulse-chase experiment in WT and KO mice islets using radioactive [$^{35}$S]-cysteine labeling. Insulin and proinsulin-related proteins were immunoprecipitated with anti-insulin polyclonal antibody coupled to Dynabeads. Densitometry of $^{35}$S-labeled insulin species was performed using a PhosphorImager. Results are representative of three experiments with three mice per group. (**C**) Kinetics of proinsulin processing in WT and KO mice islets (three biological replicates from nine mice, p = 0.006 at 180 min, t-test). (**D**) Insulin/proinsulin ratio during proinsulin processing in WT and KO mice islets (three biological replicates from nine mice, p = 0.006 at 180 min, t-test). (**E**) Western blotting with anti-HID-1 polyclonal antibody and anti-insulin polyclonal antibody in WT and HID-1 KO rat INS-1 cells. Absence of HID-1 confirms the KO effect. Proinsulin is significantly higher in HID-1 KO cells than in WT mice. (**F**) Kinetics of proinsulin1 processing analyzed with pulse-chase experiments followed by LC-MS quantification using stable isotopic [$^{13}$C,$^{15}$N]-isoleucine labeling in WT and HID-1 KO rat INS-1 cells (WT, n = 4; KO, n = 6, p = 0.008, <0.001 and 0.002 at 50, 100, 150 min, respectively, t-test). (**G**) Kinetics of insulin1 production in WT and HID-1 KO rat INS-1 cells (p = 0.005, 0.036 and 0.038 at 50, 100 and 150 min, respectively, t-test). (**H**) Insulin1/proinsulin1 ratio during proinsulin processing in WT and HID-1 KO rat INS-1 cells (p = 0.01, 0.01, and 0.019 at 50, 100 and 150 min, respectively, Mann-Whitney U Test).

The following figure supplements are available for figure 3:

**Figure supplement 1.** Quantification of proinsulin and its processing products by LC-MS in INS-1 cells.

**Figure supplement 2.** HID-1 deficiency causes a slight decrease in processing enzyme levels.

(*Steiner et al., 2016*). In pancreatic β cells, PC2 and PC1/3 are present in the SGs, and these enzymes are believed to cooperate in the processing of proinsulin. It has been suggested that PC1/3 first acts at the B–C junction to produce the intermediate des-31,32 proinsulin, whereas PC2 preferentially cleaves at the A–C junction to produce the intermediate des-64,65 proinsulin (*Halban and Irminger, 1994*). To further confirm the involvement of HID-1 in proinsulin processing and to understand which cleavage steps are affected in the absence of HID-1, we knocked out HID-1 in the insulin-secreting cell line (INS-1) using TALEN technology. As shown in *Figure 3E*, western blotting confirmed the complete absence of HID-1 protein, increased proinsulin and reduced insulin in HID-1 KO INS-1 cells.

Rodent insulins are produced from two independent genes called *Ins1* and *Ins2*. As western blotting and autoradiography of proteins labeled with [$^{35}$S]-cysteine could not detect the intermediates (des-31,32 and des-64,65) or distinguish Insulin1 from Insulin2, we employed liquid chromatography-mass spectrometry (LC-MS) to detect stable isotope-labeled proinsulin and its intermediates. Insulin1 was analyzed because products from the *Ins1* gene are predominant in rats (*Gishizky and Grodsky, 1987*; *Michael et al., 1987*) (*Figure 3—figure supplement 1A–C*). As shown in *Figure 3F and 3G*, LC-MS detected slower processing of Proinsulin1 and delayed production of mature Insulin1 in the KO INS-1 cells. The insulin/proinsulin ratio was significantly lower in the KO cells than in WT controls during the chase period (*Figure 3H*). These results were in accordance with the phenomenon observed in Hid1-betaKO mice. The retarded appearance of both des-31,32 and des-64,65 Proinsulin1 during the chase period indicated the concurrent impairment of PC1/3 and PC2 function in the KO INS-1 cells (*Figure 3—figure supplement 1D,E*).

Next we analyzed PC1/3, PC2 and carboxypeptidase E (CPE) expression in WT and KO islets. CPE is an enzyme that catalyzes the release of C-terminal arginine or lysine residues from polypeptides. A typical result from three independent experiments is shown in *Figure 3—figure supplement 2A*. Quantification showed that HID-1 KO caused a slight decrease in PC1/3 and PC2 levels, but not in CPE levels (*Figure 3—figure supplement 2B*). We further quantified immunofluorescence-labeled PC1/3 and PC2 levels in single β cells using flow cytometry (*Figure 3—figure supplement 2C*). Consistent with the above results, PC1/3 and PC2 levels were slightly lower in HID-1 KO β cells than in WT β cells. A previous study demonstrated that the 50% reduction in PC1/3 only elevated serum proinsulin/insulin ratio to 0.12 (*Zhu et al., 2002*), hence, the slight decrease in PC1/3 and PC2 levels that we observed is not sufficient to explain the drastic defect in proinsulin processing (proinsulin/insulin ratio of 0.39).

## HID-1 deficiency increases the number of ISGs

Our results demonstrated that HID-1 deficiency caused defective proinsulin processing and proinsulin accumulation in β cells. Consequently, we employed immunofluorescence imaging to reveal the subcellular localization of proinsulin. Interestingly, we observed a dispersed distribution of proinsulin throughout the HID-1 KO cells, which was unlike the perinuclear localization seen in WT cells (*Figure 4A*, see also *Figure 1—figure supplement 2B*). The density of proinsulin vesicles was significantly increased from $0.58 \pm 0.10/\mu m^3$ in WT cells to $0.99 \pm 0.07/\mu m^3$ in KO cells (*Figure 4B*). Since proinsulin is a specific ISG marker, the increase in proinsulin signals may reflect an increase in the number of ISGs. To test this hypothesis, we further checked other ISG markers, VAMP4 and Syt-IV. As shown in *Figure 4—figure supplement 1A and C*, both VAMP4 and Syt-IV localized to the perinuclear region, and co-localized with proinsulin puncta to a similar extent (similar Pearson coefficient). In HID-1 KO β cells, proinsulin and Syt-IV were similarly dispersed throughout the cytosol and displayed a similar extent of co-localization as in WT cells (*Figure 4—figure supplement 1* A and C, bottom, B and D). By contrast, VAMP4 remained in the peri-nuclear region and did not co-localize with proinsulin in KO cells (*Figure 4—figure supplement 1* A and B). Thus, proinsulin-positive ISGs in HID-1 KO cells seem to retain Syt-IV but not VAMP4, suggesting that VAMP4 recruitment to r retention in ISG depends on HID-1. Besides, a previous study has suggested that VAMP4 is not an essential component for ISG–ISG fusion process (*Wendler et al., 2001*).

To further verify the increase of ISGs, we employed electron microscopy (EM). Previous ultrastructural studies of insulin-containing granules mainly employed transmission electron microscopy (TEM) on ultrathin sections (<100 nm) (*Orci, 1976*). As illustrated in *Figure 4—figure supplement 2A*, ultrathin sectioning of granules of this size gives rise to various sub-maximum cross-sections, depending on where the section is on the z-axis, which leads to under-estimation of the granule size

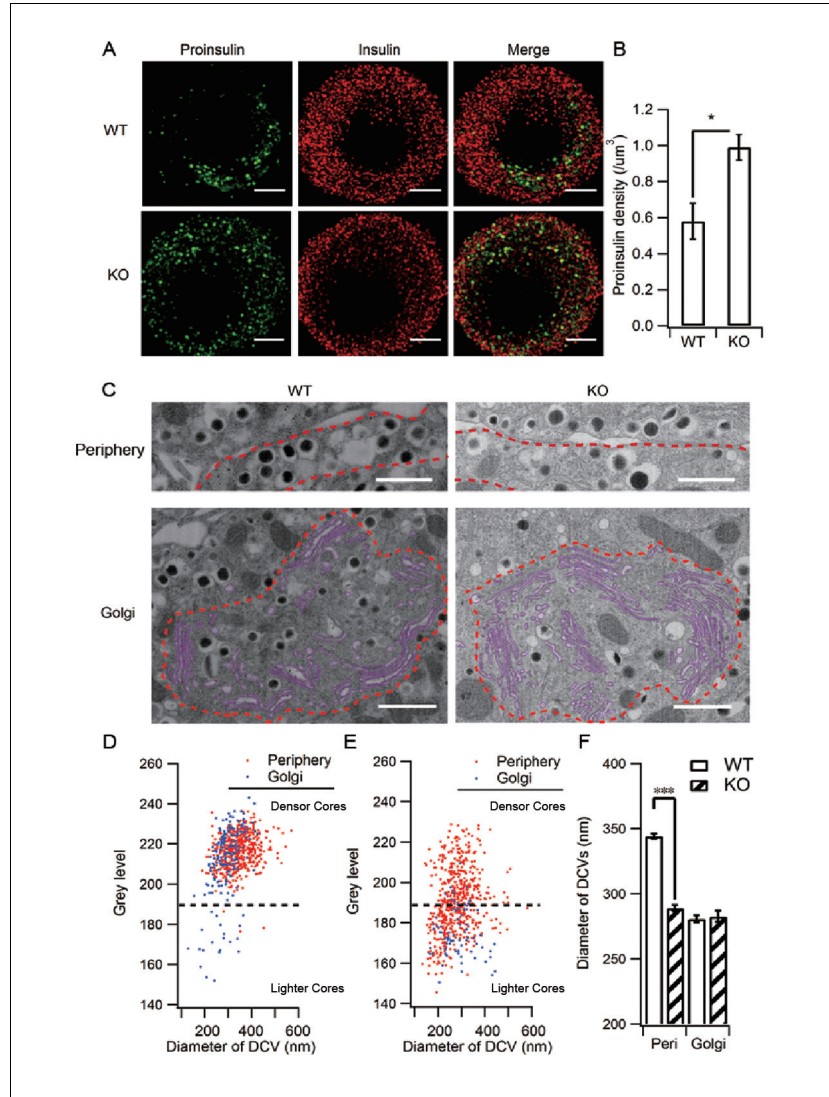

**Figure 4.** HID-1 deficiency increases the number of ISGs in β cells. Immunofluorescence analysis of ISGs (labeled with proinsulin antibody) and MSGs (labeled with insulin antibody), showing (**A**) subcellular distribution and (**B**) density in WT (n = 8) and KO (n = 14) mice (three biological replicates from six mice, p = 0.015, Mann-Whitney *U* test). Scale bar, 2 μm. (**C**) Comparison of SG morphology between WT and KO β cells. There are many small and light SGs in the peripheral region (indicated by a dashed red line, upper panel) of KO cells. Golgi stacks (highlighted in purple) in β cells form an enclosed region, which is defined as the Golgi region (indicated by a dashed red line, lower panel). Scale bar, 1 μm. (**D**) Two-dimensional (2D) scatter plots of grey level against granule size in WT β cells (Golgi – n = 179, 21 denser cores; periphery – n = 517, four lighter cores). (**E**) 2D scatter plots of grey level against granule size in KO β cells (Golgi – n = 73, 65 lighter cores; periphery – n = 527, 252 lighter cores). Each point represents a single granule. The granules calculated were from the same reconstructed volume for both WT and KO to avoid artifacts caused by staining and SEM imaging. (**F**) Granules in the periphery (Peri) region of WT β cells are significantly larger than those from KO mice (344.0 ± 2.2 nm, n = 1,070; 289.1 ± 2.3 nm, n = 1,387; respectively, p < 0.001, Kolmogorov-Smirnov test). No significant difference in granules from Golgi regions was observed between WT and KO mice (280.7 ± 2.8 nm, n = 443, respectively; 282.7 ± 4.2 nm, n = 256, respectively; p = 0.756, Kolmogorov-Smirnov test).

The following figure supplements are available for figure 4:

**Figure supplement 1.** Depletion of HID-1 leads to reduced co-localization with VAMP4, but not with Syt-IV.

**Figure supplement 2.** Ultrastructure analysis of SGs by large volume FIB/SEM.

and makes it difficult to analyze the density and detailed morphology of the granules. The recent development of combining scanning EM (SEM) with focused ion beam (FIB) (*Figure 4—figure supplement 2B*) has provided a powerful new method to study the morphology and distribution of SGs in whole cells at nanometer resolution (*Figure 4—figure supplement 2C*, *Video 1,2*). Typical SGs exhibit a dark (electron dense) central core and a delimiting membrane, separated by a clear region (halo) (*Figure 4C*). The grey levels of the dense core indicate the extent of packaging and condensation of insulin. It has been suggested that ISGs exhibit lighter dense cores than MSGs (*Furuta et al., 1998*; *Wijesekara et al., 2010*). Since the grey levels of dense cores are not directly comparable between different SEM experiments, we compared the grey level distribution between the Golgi region and the peripheral region (<1 µm from the PM) from the same SEM volume (13.7 x 11.8 x 5 µm$^3$) in both WT and KO β cells (*Figure 4C–E*). Our results showed that in WT β cells, the grey levels of dense cores are mostly distributed in a zone with higher values (grey level > 190, designated as denser cores in *Figure 4D*) and only a small proportion are in a zone with lower values (grey level < 190, designated as lighter cores in *Figure 4D*). The lighter core SGs were mostly from the Golgi region (84.0%), while the peripheral SGs are almost denser cores (99.2%). This result confirms the existence of at least two populations of SGs, ISGs and MSGs in resting WT β cells, and suggests that only a small fraction of ISGs are in the Golgi region. However, in KO cells, the clear distinction between the two populations was replaced by continuous distribution of grey levels from light to dense values (*Figure 4E*). Adopting the same separation of grey level as for WT cells, we found that there were many more granules with a lighter dense core (<190) in KO cells than in WT cells for both peripheral (47.6% vs 0.8%) and Golgi (87.8% vs 11.7%) regions, suggesting the appearance of more ISGs in both regions in the absence of HID-1. We further compared SG size and found that the SGs in peripheral regions were significantly larger than those in the Golgi region in WT β cells (*Figure 4F* and *Figure 4—figure supplement 2D,E*). The presence of larger SGs in the peripheral regions is consistent with the maturation of SGs on their way from the Golgi to the periphery because it has been reported that ISGs can fuse with each other and become larger in size during maturation (*Urbe et al., 1998*). However, HID-1 deficiency significantly decreased the size of SGs in the peripheral region (*Figure 4F* and *Figure 4—figure supplement 2F,G*), suggesting abnormal maturation of SGs.

## HID-1 deficiency blocks homotypic fusion of ISGs

The above immunofluorescence and EM data clearly demonstrated the accumulation of smaller, less dense ISGs in the absence of HID-1, which could be caused by a blockage in the homotypic fusion of ISGs (*Noske et al., 2008*; *Wijesekara et al., 2010*). To test this hypothesis, we took advantage of the high three-dimensional (3D) resolution of FIB-SEM to directly visualize homotypic fusion events, which are characterized by the appearance of two dense cores within one limiting membrane (*Figure 5A*). We observed a low frequency of homotypic fusion events (8.2 ± 0.8 per 1000 SGs) in WT β cells, mostly near the Golgi region, which was significantly reduced by ~ 8 fold (1.2 ± 0.4 per 1000 SGs) in HID-1 KO cells (*Figure 5B*).

The low probability of capturing homotypic fusion events is consistent with our notion that only a small fraction of SGs are immature (*Figure 4D*) in WT β cells. In order to integrate fusion events over time and provide independent evidence, we optimized an in vitro assay to quantify homotypic fusion (*Brandhorst et al., 2006*) (*Figure 5C*). By inserting green fluorescent protein (GFP) into the C-peptide of insulin, we labeled SGs with different colors. We verified that these GFP-labeled insulins were correctly sorted to SGs as they co-localized with chromogranin A (*Figure 5—figure supplement 1*), a marker of insulin-containing SGs. A subcellular fractionation method was employed to purify ISGs as previously described (*Chen et al., 2015*). As shown

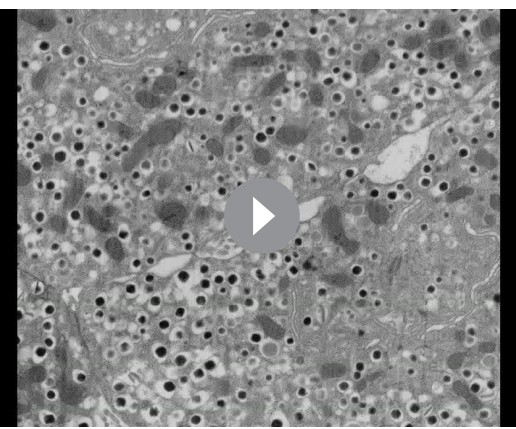

**Video 1.** The morphology of SGs in WT mouse β cells.

in *Figure 5D*, WT ISGs exhibited more fusion events, which were characterized by co-localization of the centers of bright fluorescent dots in both green and red channels (*Figure 5E*). About 7.7% of ISGs derived from WT cells underwent fusion, while the fraction decreased to 0.89% for ISGs derived from HID-1 KO cells, which was as low as that for negative controls in which ATP was depleted or the reaction system was placed on ice (*Figure 5F*). Taken together, both EM and fluorescence co-localization assays support the conclusion that HID-1 deficiency blocks homotypic fusion of ISGs.

## Granule acidification is blocked in HID-1 KO cells

It has been proposed that blockade of homotypic fusion could impair prohormone processing by inhibiting ISG acidification (*Ahras et al., 2006*). We thus analyzed intragranular acidification of SGs by employing the acidotrophic agent 3-[2,4-dinitroanilino]-3'-amino-N-methyldipropyramine (DAMP), which accumulates in secretory granules and acts as an indicator of intragranular pH (*Anderson et al., 1984*; *Orci et al., 1986*). We verified that DAMP accumulated in insulin-containing SGs by immunofluorescence co-localization (*Figure 6A*). The granular DAMP intensities displayed a significant shift to smaller values in HID-1 KO β cells compared with WT cells (*Figure 6B*), suggesting that HID-1 deficiency blocks acidification at the single granule level. We further analyzed DAMP levels in insulin-positive β cells by immunofluorescence flow cytometry. DAMP levels in HID-1 KO β cells were significantly reduced to an average of 57.3% of those in WT β-cells (*Figure 6C*). This defect was confirmed by employing another acidotropic probe (*Figure 6—figure supplement 1*), Lysosensor, which has been used to quantify intragranular pH recently (*Liu et al., 2015*; *Thevenod, 2002*). These data suggest that SG acidification is disturbed in the absence of HID-1.

## Discussion

SGs are specialized organelles that emerged during evolution for the storage and transport of regulated secretory proteins, i.e., hormones and neuropeptides (*Kim et al., 2006*). While much effort has been spent on understanding the machinery involved in the control of SG exocytosis, less is known regarding how SGs are generated from the TGN and how they mature thereafter.

### Homotypic fusion in the SG maturation process

Early morphological studies showed that newly generated ISGs are small as they emerge from the TGN, and then appear to fuse together to form larger MSGs (*Salpeter and Farquhar, 1981*; *Smith and Farquhar, 1966*; *Tooze et al., 1991*). Later, this concept of homotypic fusion of ISGs was validated using in vitro reconstitution assays (*Tooze and Huttner, 1990*; *Urbe et al., 1998*). It is believed that homotypic ISG fusion is important for the condensation and refinement of the contents and for remodeling of the membrane. However, homotypic ISG fusion has been reported in neuroendocrine cells, including PC12 cells (*Ahras et al., 2006*; *Tooze et al., 2001*; *Wendler et al., 2001*), mammotroph pituitary cells (*Farquhar and Palade, 1981*) and mast cells (*Hammel and Meilijson, 2015*), and in yeast (*Asensio et al., 2013*), but not in endocrine cells such as pancreatic β cells. Evidence for an absence of homotypic fusion in β cells mainly rests on the analysis of granule size. First, a dominant-negative construct against STX6 caused a delay in proinsulin processing but failed to affect the size of insulin granules (*Kuliawat et al., 2004*). Second, EM data in β cells did not reveal an increase in diameter of MSGs, but rather some shrinkage of granules as they age (*Hutton, 1989*; *Noske et al., 2008*). Determination of the granule size on thin sections can be tricky because planar sectioning at random depths through spherical granules tends to underestimate the diameter of granules and largely increase the variation. Besides, the study by *Noske et al. (2008)* did not compare the size of ISG and MSG, but rather reported a size change of older granules between two cells. Hence, large-volume 3D EM analysis of intact granules from multiple cells is needed for unbiased estimation of granule size and for comparison between ISGs and MSGs. In this study, we employed automatic FIB-SEM to visualize a large volume (13.7 x 11.8 x 3~8 μm$^3$) at nanometer resolution and analyzed thousands of intact granules from 15 cells. We analyzed not only the diameter, but also the grey level of the cores and the subcellular localization of granules. This comprehensive analysis enables us to separate ISGs from MSGs clearly and to conclude that MSGs have larger size in β cells (*Figure 4*). Moreover, we have directly visualized homotypic fusion events (*Figure 5A*) despite their rareness. Together with data from in vitro ISG fusion assay (*Figure 5F*), these lines of evidence clearly demonstrate the existence of homotypic ISG fusion in endocrine β cells.

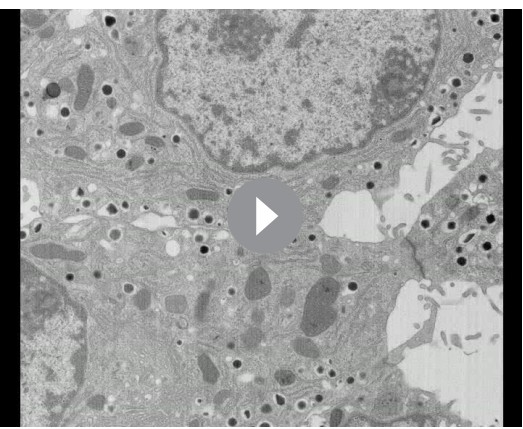

**Video 2.** The morphology of SGs in HID-1 KO mouse β cells.

Please note that maturation of SGs may not always come with an increase in diameter, MSGs can either become smaller or larger than ISGs (*Tooze et al., 2001*). Maturation involves membrane remodeling steps of both homotypic fusion and budding events. Homotypic fusion would result in an increase in the size of ISGs, whereas budding of clathrin-coated vesicles from ISG would allow removal of excessive membrane and reduction in size (*Arvan and Castle, 1998*). Hence, the size of MSGs is determined by the coordinated action of fusion and budding events and is likely to be a cell-specific property.

The search for molecules that are involved in homotypic ISG fusion has led to the identification of NSF and α-SNAP (*Urbe et al., 1998*), which promote membrane fusion by disassembling cis-SNAREs to form fusogenic trans-SNARE complexes between donor and target membranes. Later, the SNARE protein STX6, which localizes to ISGs but not MSGs, was found to be required for homotypic ISG fusion (*Wendler et al., 2001*). SM proteins are essential partners for SNARE proteins in fusion – without one or the other, no fusion occurs physiologically (*Sudhof and Rothman, 2009*). The interaction between Munc18-1, a neuronal SM protein, and syntaxin 1 is critical for the docking and priming of synaptic vesicles, which also requires Munc13 proteins. Munc13s, and related neuronal proteins called CAPS (calcium-dependent activator protein for secretion), are essential players during priming, and contain a large module called the MUN domain. It has been proposed that the MUN domain dramatically accelerates the transition from the closed syntaxin 1–Munc18-1 complex to the SNARE complex, probably by extracting the SNARE motif of syntaxin 1 from the closed conformation and providing a template for SNARE complex assembly (*Ma et al., 2011*; *Yang et al., 2015*). Structural analyses have shown that the MUN domain has a similar architecture to domains found in vesicle tethering factors such as Sec6, Tip20, and Exo70 (*Yang et al., 2015*). As such, Munc13s and CAPS belong to a family named CATCHR (complex associated with tethering containing helical rods) (*Rizo and Sudhof, 2012*; *Yu and Hughson, 2010*). CATCHR proteins have general function(s) in vesicle fusion by providing a physical link between donor and target membranes (sometimes called docking and sometimes called tethering) (*Yu and Hughson, 2010*). Moreover, as described above for Munc13-1 and for the HOPS complex, tethering factors bind to SNARE proteins and can accelerate SNARE complex assembly (*Ren et al., 2009*; *Wickner, 2010*). These findings suggest that the docking and assembly of the SNARE complex may be mediated by similar but not identical mechanisms in different systems. As HID-1 localizes to ISGs, it will be intriguing to test whether HID-1 can serve as a tethering factor to bridge disparate ISG membrane or as a priming factor for SNARE complex assembly. The large volume SEM and in vitro fusion assays we developed here provide the first evidence of ISG homotypic fusion during the maturation of insulin granules.

## Insulin processing and granule acidification

Nascent ISGs generated from the TGN contain proinsulin, which does not efficiently pack into the higher-order granule matrix (*Steiner, 1973*). However, processing of proinsulin to insulin and C-peptide by the endoproteases PC1/3 and PC2 during ISG maturation allows insulin to condense (*Arvan et al., 1991*). In the current study, we show for the first time that HID-1 KO impairs proinsulin processing (*Figures 2* and *3*). This answers the long-standing question of why HID-1 deficiency does not block DCV exocytosis (*Mesa et al., 2011*) (*Figure 1—figure supplement 3*) but causes defective insulin signaling in both mouse and worm (*Ailion and Thomas, 2003*).

Interestingly, blockade of ISG homotypic fusion by targeting Syt-IV or STX6 slowed down prohormone processing (*Ahras et al., 2006*; *Wendler et al., 2001*). It is not clear how homotypic fusion affects proinsulin processing. One possibility is that the processing enzymes, PC1/3 and PC2, are mis-sorted to lysosomes for degradation, as suggested for Rab-2 (*Edwards et al., 2009*). However,

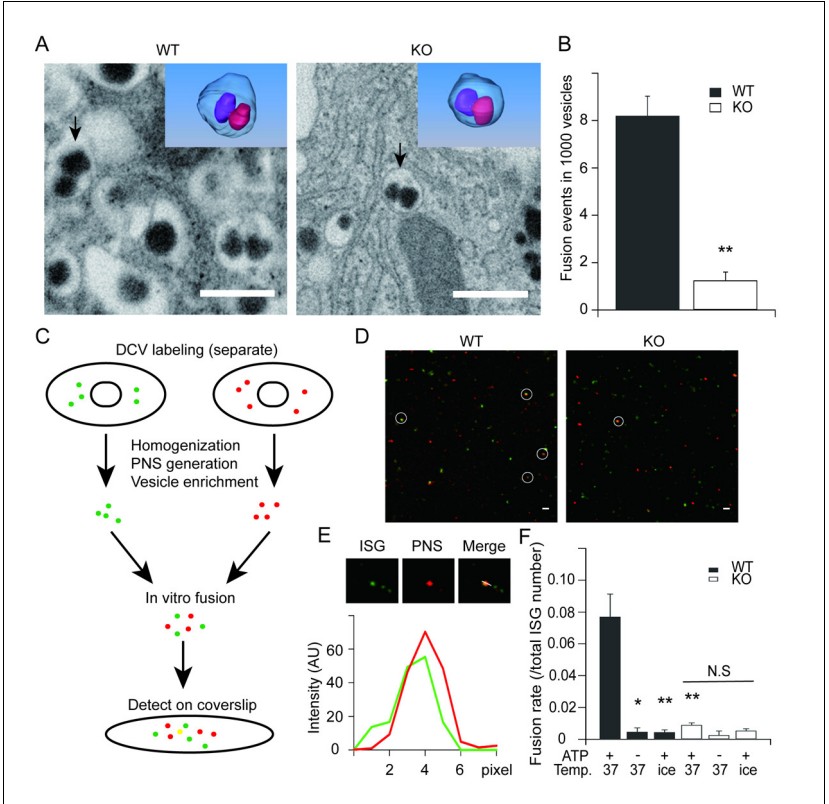

**Figure 5.** HID-1 deficiency disrupts homotypic fusion of ISGs. (**A**) Representative sections from FIB/SEM reconstructions containing ISG fusion events. The arrowed fusion events are segmented and rendered in 3D (top corner). Scale bar, 500 nm. (**B**) Comparison of the frequency of fusion events. There were 8.2 ± 0.8 fusion events per 1,000 SGs in WT (19/2632, 4/532, 22/2253, n = 3 cells) and 1.2 ± 0.4 in KO (3/3024, 1/1043, 3/1539, n = 3 cells), p = 0.002, t-test. (**C**) An overview flowchart of the in vitro fusion assay. (**D**) Representative images of fusion reactions using either WT cells or HID-1 KO INS-1 cells. Fusion events are marked with white circles. Scale bar, 3 μm. (**E**) A typical fusion event as shown in panel (**D**). The graph shows the fluorescence intensity profiles of green SGs (ISG) and red SGs (PNS). (**F**) ISGs and PNS obtained from both WT and HID-1 KO INS-1 cells were subjected to in vitro fusion assays (n = 3) with or without ATP at 37°C or on ice. The fusion rate was calculated by dividing the number of fusion events by the total number of ISGs. (p = 0.03, 0.007 and 0.009 for $WT_{37°C-ATP}$, $WT_{ice+ATP}$ and $KO_{37°C+ATP}$ vs $WT_{37°C+ATP}$, respectively, t-test. N.S = not significant.)

The following figure supplement is available for figure 5:

**Figure supplement 1.** Insulin-EGFP protein localizes to SGs.

PC1/3 and PC2 levels in the HID-1 KO β cells were only slightly decreased (*Figure 3—figure supplement 2*), and this small change is not sufficient to explain the drastic blockade of proinsulin processing. Another possibility is that the endopeptidase activities of PCs are affected in the absence of HID-1. It has been demonstrated that acidification is one of the most important steps in proinsulin processing and granule maturation (*Orci et al., 1986*). Specifically, PC1/3 and PC2, which process proinsulin, are strictly pH-dependent (*Lamango et al., 1999*; *Orci et al., 1994*; *Seidah et al., 1993*; *Smeekens et al., 1992*; *Steiner, 1998*). Indeed, we have provided the first evidence that granule acidification is blocked in the absence of HID-1 (*Figure 6*). Hence, it is plausible that HID-1 KO inhibits ISG homotypic fusion and thereby blocks granule acidification, which impairs the pH-dependent activity of PCs and retards proinsulin processing.

Progressive granule acidification is driven by the vacuolar $H^+$-ATPase, which transports protons into the lumen and causes a decrease of the intragranular pH (*Jefferies et al., 2008*). We verified that vacuolar $H^+$-ATPase was present on ISGs as well as on MSGs (*Figure 6—figure supplement 1*).

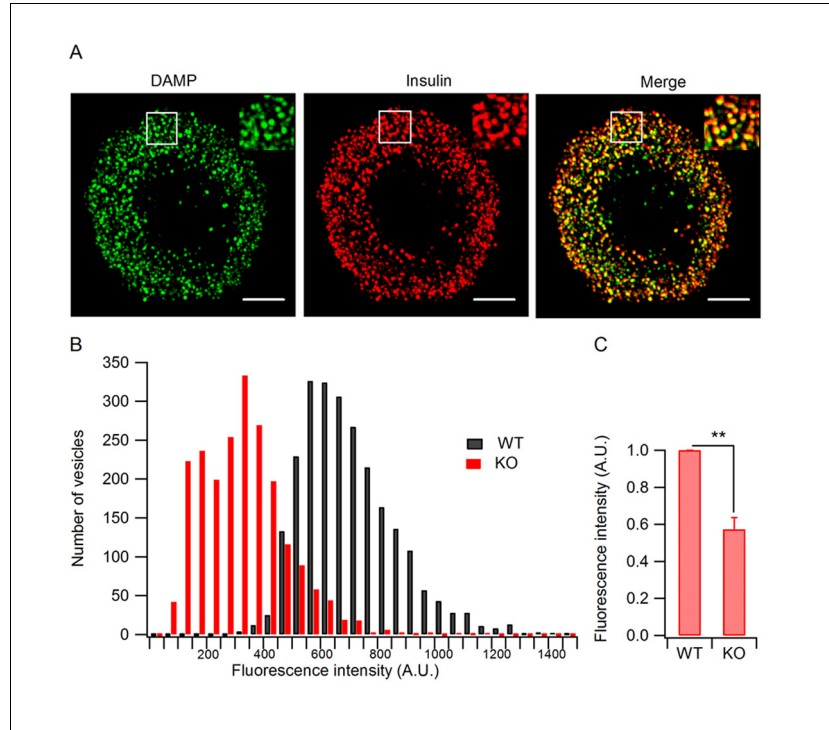

**Figure 6.** Granule acidification is blocked in HID-1 KO cells. Indirect immunofluorescence was performed on dispersed β islet cells, after 1 hr of incubation with 30 µM DAMP in Hanks balanced salt solutions (4 mM glucose). Anti-DNP was used to detect DAMP. Anti-insulin was used to label insulin-containing granules. (**A**) 3D-SIM images of WT β cells, showing colocalization of DAMP (green) with insulin granules (red). Scale bar, 2 µm. (**B**) Comparison of DAMP intensity between WT (n = 2443) and KO (n = 2117) insulin granules (imaged by confocal microscopy, 50 cells for each group and three biological replicates with six mice). The difference is significant (p < 0.001, Kolmogorov-Smirnov test). (**C**) Mean fluorescence intensity of DAMP from insulin-immunoreactive cells was analyzed by flow cytometry (four biological replicates with eight mice, p = 0.015, Wilcoxon test).

The following figure supplement is available for figure 6:

**Figure supplement 1.** Verification of the granule acidification defect in HID-1 KO cells and the granular location of H⁺-ATPase.

Although it is not clear how homotypic ISG fusion causes granule acidification, it has been proposed that ISG–ISG fusion could increase the density of the vacuolar H⁺-ATPase, as well as decreasing the H⁺ permeability (*Wu et al., 2001*). We propose that homotypic ISG fusion plays a pivotal role during SG maturation by condensing not only prohormones but also membrane machinery components, such as vacuolar H⁺-ATPase, which are required for the acidification of prohormones and their processing to form active peptides. Blockade of homotypic ISG fusion arrests SGs in an immature state and impairs subsequent maturation steps such as acidification, processing and sorting.

## Physiological and pathophysiological implications

As early as 1972, Duckworth and Kitabchi reported that obese type 2 diabetic patients exhibited a greater rise in the proinsulin to insulin ratio than non-diabetic controls after an oral glucose challenge (*Duckworth and Kitabchi, 1972*). This result was subsequently confirmed in several stages of type 2 diabetes mellitus (T2DM) (*Csorba, 1991*). In recent years, an increasing amount of experimental and clinical evidence indicates that proinsulin and proinsulin-like compounds (mainly des-31,32 proinsulin) in serum have a close relationship with T2DM, whereas hyperinsulinemia in the early and intermediate disease stages is due to cross-reactivity between insulin and its precursors (*Pfutzner et al., 2004*).

The mechanisms of hyper-proinsulinemia associated with T2DM are still poorly understood. One plausible mechanism is an impaired conversion of proinsulin to insulin, i.e., the involvement of defective processing enzymes. Indeed, genetic variants that affect proinsulin folding and processing result in susceptibility to T2DM (*Liu et al., 2010*; *Zheng et al., 2012*). In the current study, we have identified HID-1 as a novel player in homotypic ISG fusion, ablation of which impairs proinsulin processing. Apparently, many factors are required to orchestrate multiple steps of SG maturation from biogenesis at the TGN, through homotypic fusion and granule acidification, to cargo refinement. These players include, but are not limited to, Rab2, Rab2 effectors and SNARE accessory proteins. Rab2 localizes to the Golgi and is likely to be involved as a master regulator in multiple steps of vesicular trafficking, including budding from TGN, transportation, homotypic fusion, and sorting of cargoes (*Cao et al., 2013*; *Edwards et al., 2009*; *Stenmark, 2009*). The Rab2 effectors RIC-19/ICA69, RUND-1, CCCP-1 and PICK1 are implicated in the maturation of SGs in both *C. elegans* and mammalian cells (*Ailion et al., 2014*; *Holst et al., 2013*). HID-1, STX6 and Syt-IV are required for homotypic ISG fusion (*Ahras et al., 2006*; *Wendler et al., 2001*). It will be important to expand the complete list of molecular players involved in SG maturation and to understand the sequence in which they act as well as the underlying mechanisms.

As peptide hormones and neuropeptides regulate a variety of physiological processes, including development, metabolism, immunity and brain functions (*Nässel and Larhammar, 2013*), genetic defects in this pathway are likely to contribute to multiple diseases not limited to T2DM. In fact LRRK2, a major susceptibility gene for inflammatory bowel disease (IBD), was found to recruit Rab2 to ISGs for the selective sorting of lysozyme during SG maturation in Paneth cells (*Zhang et al., 2015*). In light of this and other studies (*Hatanaka et al., 2011*; *Katsuta et al., 2015*; *Stoy et al., 2007*; *Zhang et al., 2015*), *Hid1* and other genes involved in SG maturation should be considered as previously unappreciated candidates for susceptibility for T2DM, IBD and many other diseases involving peptide hormones and neuropeptides.

In summary, we have identified a novel player that is essential for homotypic fusion of ISGs. We provide evidence that homotypic ISG fusion is essential for granule acidification and cargo processing. Further biochemical and structural studies will be needed to reveal how HID-1 cooperates with SNARE proteins to mediate homotypic ISG fusion.

## Materials and methods

### Generation of the HID-1 KO mice

We first generated *Hid1*-floxed mice (*Hid1^{flox/flox}*) by homologous recombination of a targeting vector that replaced exons 2–6 of the coding sequence with the neomycin resistance gene. Embryonic stem (ES) cell clones were screened for homologous recombination by Southern blot analysis and positive clones were injected into C57BL/6J blastocysts to generate chimeras. *Hid1^{flox/flox}* mice were identified by PCR using the forward primer 5'-TCCTAATGTACCCAGTGCTCG-3' and the reverse primer 5'-CATCACCAGATCACATCCCAC-3. *Hid1^{flox/flox}* mice were bred with RIP-Cre mice to produce β-cell-specific HID-1 KO mice (Hid1-betaKO). The presence of the Cre recombinase will cause a major deletion of exons 2–6 and then a frameshift mutation with premature termination of translation of the HID-1 protein. All mouse strains used in the experiments were backcrossed onto the C57BL/6J genetic background. All experiments were approved by the Animal Care Committee at the Institute of Biophysics (License No. SYXK2016-19).

For intra-peritoneal glucose-tolerance tests (IGTT), the mice were first fasted overnight and then given an i.p. glucose injection (2 g/kg body weight). Venous blood was drawn from the tail vein, and the blood glucose level was measured using a glucometer (ACCU-CHEK Active, Roche, Basel, Switzerland). For insulin-tolerance tests (ITT), mice were first fasted for 4–5 hr and then 0.5 U/kg body weight of human regular insulin (Sigma, St. Louis, MO) was i.p. injected into each mouse. Blood glucose levels were measured with a glucometer.

### Cell line

The INS-1 cell line was obtained from the Cell Resource Center, Peking Union Medical College (the headquarter of National Infrastructure of Cell Line Resource, NSTI, Beijing, China), which was

checked free of mycoplasma contamination. The identity of INS-1 cell line was verified with parallel transcriptome studies using RNA sequencing.

## $C_m$ measurement

Conventional whole-cell recordings were performed at 30°C with Sylgard-coated 2–4 MΩ pipettes. β cells from 12-week-old male WT and Hid1-betaKO mice were isolated and cultured as described previously (*Kang et al., 2006*). For $C_m$ recording, the extracellular solution contained: 140 mM NaCl, 5.6 mM KCl, 1.2 mM MgCl$_2$, 5 mM glucose, 10 mM HEPES, 20 mM TEA-Cl, 10 mM CaCl$_2$, and 2 mM forskolin, pH 7.2, and the whole-cell pipette solution contained: 125 mM CsGlu, 5 mM NaCl, 1 mM MgCl$_2$, 2 mM MgATP, 0.3 mM GTP, 20 mM HEPES and 0.5 mM Ca$^{2+}$-EGTA, with the pH adjusted to 7.2 using CsOH or HCl. $C_m$ measurements were performed using the Lindau–Neher technique implemented as the 'sine + dc' mode of the software lock-in extension of Pulse (HEKA, Lambrecht, Germany). After establishing the whole-cell configuration, the cells were held at a resting potential of −70 mV. Depolarization trains of 1 Hz that contain ten 200 ms depolarizations to 0 mV were delivered to the cells. Finally, the $C_m$ traces were imported to IGOR Pro (WaveMetrics, Lake Oswego, OR) for further analysis.

## Antibodies

The following primary antibodies were used for immunofluorescence and western blot analysis: mouse monoclonal antibody against native insulin (Abcam, Cambridge, MA, ab7760), guinea pig polyclonal antibody against total insulin (preproinsulin, proinsulin and insulin) (Abcam, ab7842); mouse monoclonal anti-proinsulin (HyTest, Turku, Finland, CCI-3, specific for proinsulin and does not recognize insulin or C-peptide); rabbit monoclonal anti-glucagon (Abcam, ab92517); mouse mono-clonal anti-HID-1 (Origene, Rockville, MD, TA501311) or rabbit polyclonal anti-HID-1 (Proteintech, Wuhan, China, 21174-1-AP); rabbit polyclonal anti-PC1/3 (Abcam, ab3523); rabbit polyclonal anti-PC2 (Abcam, ab3533); mouse anti-CPE (BD, San Jose, CA, 610759), rabbit polyclonal anti VAMP4 (Abcam, ab3348); mouse monoclonal anti-Synaptotagmin 4 (Abcam, ab57473). The following sec-ondary antibodies were used: Alexa Fluor 488-conjugated goat anti-mouse; Alexa Fluor 555-conju-gated goat anti-rabbit; Alexa Fluor 405-conjugated goat anti-mouse (all from Thermo Fisher, Waltham, MA).

## ELISA detection of insulin secretion

For serum proinsulin and insulin detection, blood samples were collected into centrifuge tubes from the mouse orbital vein with no anti-coagulant. Then, the clotted blood was centrifuged at 2000 x g to obtain serum. The serum proinsulin and insulin concentration was detected using mouse proinsu-lin and insulin ELISA kits (ALPCO, Salem, NH, 80-PINMS-E01 and Shibayagi, Shibukawa, Japan, AKRIN-011S), respectively. For the in vitro insulin release assay, isolated islets were cultured in RPMI medium 1640 (with 7 mM glucose, GIBCO) for about 24 hr. After three washes with phosphate-buffered saline (PBS), 20 islets from each WT and KO group, were pre-incu-bated for 60 min in 4 mM glucose Krebs-Ringer bicarbonate HEPES buffer (KRBH) containing: 114 mM NaCl, 4.7 mM KCl, 1.2 mM KH$_2$PO$_4$, 1.16 mM MgSO$_4$, 0.5 mM MgCl$_2$, 2.5 mM CaCl$_2$, and 20 mM HEPES with 0.2% BSA, pH 7.4. Next, groups of islets were batch-incubated in 0.2 ml stimula-tion media (20 mM glucose in KRBH). At each measurement time point (0, 30 and 60 min), 0.1 ml incubation medium was withdrawn for insulin measurement after gentle agitation and was replaced by 0.1 ml fresh KRBB solution supplemented with 20 mM glucose. All operations were conducted under dissection microscopy to avoid damaging the islets. Proinsulin and insulin were quantified by ELISA with commercially available kits (ALPCO, 80-PINMS-E01 and Shibayagi, AKRIN-011S), respectively.

## Islet morphology analysis and immunohistochemistry

Pancreases were dissected from paired WT and Hid1-betaKO mice and fixed with 4% paraformalde-hyde in PBS for 24 hr. After dehydration, the samples were embedded in paraffin. Continuous paraf-fin sections were obtained at 80 μm intervals from each pancreas. We studied islet morphology for each section using H&E staining, and the sections were scanned with a Leica SCN400 (Leica

Biochemistry | Cell Biology

Microsystems, Wetzlar, Germany). Islet size and number of islets/mm$^2$ were determined by ImageJ (National Institutes of Health, https://imagej.nih.gov/ij/ ).

For immunohistochemistry analysis, pancreas slices or cell samples were fixed with 4% paraformaldehyde in PBS for 15 min, followed by permeabilization in PBS containing 0.5% Triton X-100 (MERCK, Billerica, MA) for 10 min and blocking in PBS containing 5% goat serum for 60 min. The samples were incubated for 60 min in PBS containing primary antibodies and 2.5% goat serum and then exposed to fluorescent dye-conjugated secondary antibodies for 60 min at 37°C.

## Fluorescence Imaging

All confocal microscope images were generated using an Olympus FV1200 Laser Scanning Confocal Microscope (Olympus, Tokyo, Japan) with a 60x (NA = 1.40) oil objective. The confocal settings used for image capture were held constant when samples were being compared. Images were quantified and analyzed using Image J software (National Institutes of Health). The total fluorescence intensity of pixels in the WT samples was set to an arbitrary value of 1.0 fluorescence unit (A.U.) to enable comparison with other test samples.

3D-SIM images were generated using a DeltaVision OMX V3 System (GE Healthcare, Little Chalfont, UK) with a 100x (NA = 1.40) oil objective. The images were reconstructed, aligned and viewed using softWoRx software (GE Healthcare). Cell volume calculation and vesicles detection were performed though Imaris software (version 8.1.3, Bitplane, Concord, MA). The Spot function in Imaris automatically located proinsulin and insulin vesicles based on size and intensity thresholds. Each vesicle was represented by a sphere of arbitrary size determined by the user. The density of proinsulin vesicle was defined as the number of proinsulin puncta per cell volume (number/$\mu m^3$). All the 3D-SIM images were shown as maximum intensity projection throughout the cell.

## Generation of HID-1 KO INS-1 cells

The INS-1 cell-based $Hid1^{-/-}$ cell line was constructed by TALEN technology following the manufacturer's instruction (SIDANSAI Biotechnology, Shanghai, China). Briefly, the sequences for targeting $Hid1$ (LA: gtggagaagctggtgcaa, NR1: ttctccttctcagagtg) were inserted into plasmids pTALEN-L and pTALEN-R, respectively, and verified by DNA sequencing. The two plasmids were co-transfected into INS-1 cells and positive clones were selected by puromycin. The HID-1 KO effect was confirmed by western blot analysis.

## Monitoring proinsulin processing in mouse islets with radioactive isotopic labeling

Islets of Langerhans were isolated and cultured overnight in RPMI-1640 medium containing 7 mM glucose and 10% fetal calf serum. After overnight culture, islets were labeled in groups of 180~200 in 20 $\mu$l of pulse medium containing 28 mM glucose and 55 $\mu$Ci [$^{35}$S]-cysteine (>1000 Ci/mmol) (PerkinElmer, Waltham, MA). After 60 min of pre-incubation at 9~11°C, islet incubations were continued at 37°C in a water bath for 1 hr (pulse). Islets were then washed, divided into four batches and incubated for 1, 2, and 3 hr in medium containing 3 mM glucose and 65 mg/L of unlabeled cysteine. After the pulse and chase incubations, the islets were washed, re-suspended in immunoprecipitation buffer (0.05 M Tris-HCl, 0.15 M NaCl, 1% Triton X-100, 1% sodium deoxycholate, 0.1% SDS, pH 7.4) containing a mixture of protease inhibitors, and lysed for 1 hr at 4°C. The supernatants, after centrifugation for 10 min at 13,000×g, were then treated with an immuno-affinity absorbent consisting of guinea pig anti-insulin polyclonal antibody coupled to Protein A Dynabeads (Invitrogen, Carlsbad, CA). Insulin and proinsulin-related immuno-reactive proteins were eluted from the beads with 20 $\mu$l of NuPAGE LDS sample buffer at 70°C for 10 min. The supernatants were then loaded on NuPAGE Bis-Tris gels (10%) (Invitrogen) with MES running buffer. The proteins were then transferred to a PVDF membrane at 50 V for 80 min. Densitometry of the $^{35}$S-labeled insulin species on the membrane was performed using a Phosphorimager (Bio-Rad, Hercules, CA).

## Monitoring proinsulin processing in rat INS-1 cells with stable isotopic labeling

INS-1 cells were cultured in RPMI-1640 supplemented with 10% FBS, 50 $\mu$M 2-mercaptoethanol, 1 mM sodium pyruvate, and penicillin/streptomycin. After starvation in isoleucine-free medium for

50 min, cells were labeled with pulse medium containing [$^{13}$C, $^{15}$N]-isoleucine (Sigma) and 28 mM glucose for 50 min at 37°C. Cells were harvested after chasing for 50, 100 and 150 min in medium containing 3 mM glucose and unlabeled isoleucine. Cells were washed and sonicated, followed by acetonitrile precipitation for mass spectrometry-based analysis of insulin species. Samples were analyzed on an LC-ESI-Q Exactive MS in MS scan mode to detect 950–1400 m/z, with mobile phases of 0.1% formic acid in water (A) and 0.1% formic acid in 90% acetonitrile (B). Peaks of proinsulin1, des-31,32 proinsulin1, des-64,65 proinsulin1 with charge 7+, and insulin1 with charge 5+ were extracted using Xcalibur 2.2 software (Thermo Fisher), and the corresponding peak areas of insulin-related peptides were calculated for quantitative analysis.

## FIB/SEM

Islets were harvested and fixed in 2.5% polyformaldehyde and glutaraldehyde in PBS with a pH of 7.4 at room temperature for 1 hr. Then, samples were post-fixed with 1% osmium tetroxide in 0.1 mol/L sodium cacodylate for 1.5 hr, and stained with 1% (w/v) uranyl acetate in double distilled water for 50 min to increase the contrast under SEM. After washing and dehydration, samples were embedded in Embed 812 and stored for use. The resin-embedded block was trimmed by a diamond knife using a Leica ultra-microtome EM UC6 (Leica) to expose the islet. The block was remounted with the exposed islet upwards and glued for use. The surface of the block was coated with a thin layer of carbon to increase the electrical conductivity, then the block was transferred into the FIB/SEM chamber.

The ultra-structural 3D study was carried out using a Helios Nanolab 600i dual-beam SEM (FEI, Hillsboro, OR), which combines high-resolution field-emission SEM with a focused gallium ion beam. A 1-μm-thick layer of platinum was deposited on the block above the region of interest to protect the specimen and to reduce FIB milling artifacts. Automated sequential FIB milling and SEM imaging were conducted as described previously (*Merchan-Perez et al., 2009*). A layer of platinum (~0.8 μm thick) was deposited on a surface perpendicular to the block face. The block face was imaged using an electron beam with an acceleration voltage of 2 keV, a current of 0.23 nA, and a dwell time of 10 μs/pixel. After the surface was imaged, a gallium ion beam with an acceleration voltage of 30 keV and a current of 0.79 nA was used to remove a 20-nm thick superficial layer from the block face for the next round of imaging and milling.

## SEM data visualization and measurements

The alignment, volume trimming, and segmentation of SEM image stacks were done by Amira (Visualization Sciences Group, FEI). After reconstruction and trimming, our final reconstructions resulted in a volume range of 13.7 x 11.8 x 3~8 μm$^3$ (voxel size: 6.7 x 6.7 x 20 nm$^3$). It has been reported that in some acinar cells, the Golgi apparatus appears like a basket (*Koga et al., 2016*; *Watanabe et al., 2012*). Our large volume 3D SEM data of Golgi stacks in β cells showed the same basket shape (*Video 3*). The basket region encircled by Golgi stacks was defined as the Golgi region. SGs localized within 1 μm of the plasma membrane PM were defined as peripheral.

We analyzed seven reconstructed volumes containing eight different cells for KO, and four reconstructed volumes containing seven different cells for WT. Data sets were collected from three different mice for both WT and KO mice. Periphery regions were selected randomly with a relative size of the golgi region in each cell. We use a semi-automatic procedure to calculate the diameter of SGs (membrane edge to membrane edge) and the grey level of dense cores as previously reported (*Marsh et al., 2001*). We manually draw a single contour around individual SGs where the area was maximal among cross sections. The area within this contour was calculated, and used to determine the diameter of an equivalent circle. Granule selection and

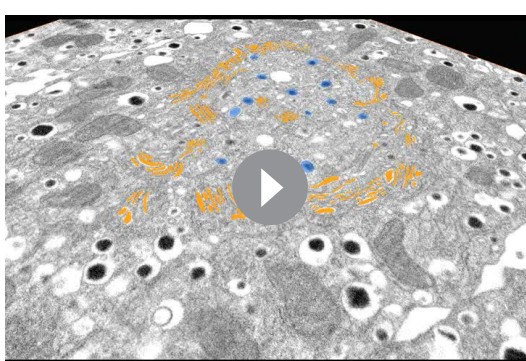

**Video 3.** 3D reconstruction of the Golgi basket in a representative WT mouse β cell.

calculation were done by three different researchers using ImageJ open source software (National Institute of Health).

The algorithm for vesicle dense core identification and segmentation was implemented in Matlab (MathWorks, Natick, MA). First, we applied a grey morphological operation to estimate the local background level and noise level. Then we subtracted the local background from the denoised image, and used a smooth function to remove noise and small particles. When the background and noise were removed, we applied auto local-maxima detection to identify the center of the vesicle dense core, and normalized the threshold for vesicle core segmentation. Finally, after applying filters to reject falsely detected results, the identification and segmentation of the dense core were completed. The grey level was calculated as the average grey level of the dense core.

## Plasmid construction and retrovirus packaging

For in vitro fusion assays, the GFP or mOrange2 sequence was inserted into the C-peptide of proinsulin and then the fusion sequence was ligated into pQCXIP vector. Retrovirus was packaged by cotransfecting plat-E cells with pIns1-EGFP or pIns1-mOrange2 and pVSV-G and pPhit using lipofectamine 2000 (Invitrogen). The retrovirus-containing supernatant was collected and filtered 48 hr post-transfection. WT or HID-1 KO INS-1 cells were infected with retrovirus in the presence of 10 μg/ml polybrene. The infected cells were selected and maintained with puromycin to generate stable cell lines.

## In vitro fusion assay

The procedure to enrich ISGs and MSGs was described in detail previously (*Chen et al., 2015*). Briefly, 8–12 plates (10 cm) of cells were collected and homogenized. The post nuclear supernatant (PNS) was acquired and subjected to sucrose density gradient centrifugation and then the ISG and MSG fractions were collected. For in vitro fusion assays, the ISGs from GFP-labeled cells were incubated with PNS from mOrange2-labeled cells in the presence of 11.25 mM HEPES, 1.35 mM Mg(Ac)$_2$, 0.18 mM DTT, 45 mM KAc, 1.5 mg/ml WT or KO INS-1 cytosol, and an ATP-regenerating system containing 3.3 mM ATP, 26.7 mM creatine phosphate and 107 U/ml creatine kinase or an ATP-depletion system containing 150 U/ml hexokinase and 25 mM glucose. The fusion reaction was performed at 37°C with slow agitation for 45 min, while the control replicate was left on ice. 20 μl of the samples were pipetted onto coverslips and detected with confocal microscopy (FV1200, OLYMPUS) following a short delay to allow precipitation. For each reaction, about 20 random fields were imaged and the acquired images were analyzed. A co-localized dot would be counted as a fusion event when the distance between the fluorescent centers of the two channels was less than 1 pixel and the difference between the fluorescent intensity of two channels was less than 3fold to eliminate potential crosstalk.

## Western blot assay

The cells or islet samples were lysed in standard RIPA buffer (150 mM NaCl, 1% NP-40, 0.5% deoxycholic acid, 0.1% SDS, and 50 mM Tris, pH 8.0), and the centrifuged supernatant was adjusted to same total protein concentration following protein quantification by a standard BCA method. Western blot analysis of protein expression were carried out as previously described (*Wang et al., 2011*). At least three independent western blots were conducted, and one typical blot is presented.

## Flow cytometry analysis

Isolated mouse islets were digested into single cells by trypsin. 100 μl of cell suspension (~$10^6$ cells) was fixed and permeabilized for immunostaining and flow cytometry analysis. Cells were incubated with primary antibodies against PC1/3 or PC2, and antibody against insulin to indicate β cells. Then the secondary antibodies were added. Cells were examined using a flow cytometer (BD Influx), and data were analyzed using Flow Jo software (Treestar, Ashland, OR).

## Intragranular pH assay

The intragranular pH assay was carried out as described previously (*Louagie et al., 2008*; *Orci et al., 1986*). In brief, dispersed islet cells were pre-incubated for 1 hr in KRBB solution. Then 30 μM DAMP (Molecular Probes, Eugene, OR) was added to the medium for 1 hr. Subsequently,

cells were fixed with 4% paraformaldehyde in phosphate-buffered saline (PBS; pH 7.4) followed by immunostaining and flow cytometry analysis as described above. Anti-DNP-KLH secondary antibody (Molecular Probes) was used for fluorescence microscopy. Anti-insulin antibody was used to label β cells.

## Data analysis

Data analysis was conducted using IGOR Pro 5.01 (Wavemetrics) unless otherwise stated. The data are presented as the mean value ± S.E.M. of the indicated number of sample size. The sample sizes were determined by the reproducibility of the experiments and are similar to those generally employed in the field. Statistical significance in two-way comparisons was determined by a Student's t-test when the data meet the normal distribution, whereas ANOVA analysis was used when comparing more than two datasets. If the data did not meet the normal distribution, the Wilcoxon test, Mann-Whitney test, or Kolmogorov-Smirnov test was used. The statistical test and significance level are indicated in the figure legends. Asterisks denote statistically significant differences from the control. (*) $p < 0.05$; (**) $p < 0.01$; and (***) $p < 0.001$.

## Acknowledgements

We would like to thank the Center for Biological Imaging (CBI), Institute of Biophysics and Chinese Academy of Science for FIB/SEM and SIM work. We are grateful to Ms. Shuoguo Li for her excellent help with SIM imaging; Dr. Wei Ji, Dr. Xiang Zhang and Dr. Bei Liu for their excellent help with super-resolution imaging and data analysis; Ms. Yan Teng for her technical support with confocal imaging; Ms. Junfeng Hao for her technical support for pancreatic section and H&E staining; Dr. Jianguo Zhang for operating FIB/SEM; and Dr. Junying Jia and Dr. Chunchun Liu for their technical support with flow cytometry. The *Hid1* conditional knockout mice were obtained from the Model Animal Research Center of Nanjing University (Nanjing, PR China).

## Additional information

### Funding

| Funder | Grant reference number | Author |
| --- | --- | --- |
| Ministry of Science and Technology of the People's Republic of China | 2016YFA0500200 | Tao Xu |
| Ministry of Science and Technology of the People's Republic of China | 2013CB910103 | Pingyong Xu |
| National Natural Science Foundation of China | 31130065 | Tao Xu |
| National Natural Science Foundation of China | 31127901 | Tao Xu |
| National Natural Science Foundation of China | 31127002 | Tao Xu |
| National Natural Science Foundation of China | 31400658 | Wen Du |
| National Natural Science Foundation of China | 31300700 | Dongwan Cheng |

The funders had no role in study design, data collection and interpretation, or the decision to submit the work for publication.

### Author contributions

WD, MZ, Performed most of the experiments, Acquisition of data, Analysis and interpretation of data, Drafting or revising the article, Contributed unpublished essential data or reagents; WZ, Analyzed the morphology of the SGs by electron tomography, Acquisition of data, Analysis and interpretation of data, Drafting or revising the article, Contributed unpublished essential data or reagents;

DC, Akinetics of proinsulin processing, Acquisition of data, Analysis and interpretation of data, Drafting or revising the article; LW, Performed the immunostaining of islets, Acquisition of data, Analysis and interpretation of data; JL, In charge of cell culture, transfection and selection of stable cell lines, Acquisition of data; ES, Participated in the design of the experiments and discussion of the results; WF, Helped with expression of the HID-1 protein and participated in extensive discussions during the experiments, Drafting or revising the article; YX, Analyzed the morphology of the SGs by electron tomography, Acquisition of data, Analysis and interpretation of data, Contributed unpublished essential data or reagents; PX, Performed partial experimental work and participated in the design of the experiments and the writing of the paper, Conception and design, Analysis and interpretation of data, Drafting or revising the article; TX, Planned and supervised the experiments, Performed data analysis, Arranged the figures and wrote the manuscript, Conception and design, Analysis and interpretation of data, Drafting or revising the article

### Author ORCIDs

Tao Xu, http://orcid.org/0000-0002-8260-9754

### Ethics

Animal experimentation: This study was performed in strict accordance with the recommendations in the Guide for the Care and Use of Laboratory Animals of Institute of Biophysics, Chinese Academy of Sciences. All of the animals were handled according to approved institutional animal care and use committee (IACUC) protocols of the Institute of Biophysics. All experiments were approved by the Animal Care Committee at the Institute of Biophysics (license number: SYXK2016-19). All surgery was performed under sodium pentobarbital anesthesia, and every effort was made to minimize suffering.

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
