## [Decision Letter]

Thank you for submitting your article "HID-1 is required for homotypic fusion of immature secretory granules during maturation" for consideration by *eLife*. Your article has been reviewed by three peer reviewers, including Edwin R Chapman (Reviewer #1) and Jens Rettig (Reviewer #2), and the evaluation has been overseen by a Reviewing Editor and Richard Aldrich as the Senior Editor.

The reviewers have discussed the reviews with one another and the Reviewing Editor has drafted this decision to help you prepare a revised submission.

Summary:

The manuscript by Du et al. investigates the role of the HID-1 protein in the biogenesis of large dense-core granules (LDCVs) in pancreatic β cells. The authors and the Nonet lab have shown previously that HID-1 localizes to the intracellular membranes like Golgi apparatus and dense-core vesicles through myristoylation at its N-terminus (Mesa et al., 2001; Wang et al., 2011; Yu et al., 2011). It was proposed that HID-1 is involved in some step of LDCV biogenesis, maturation or trafficking, however, which step is affected remained unclear. In this manuscript Du et al. generate a conditional knockout of HID-1 and show in a very elegant series of experiments that HID-1 is required for homotypic fusion of immature LDCV precursors. As a result, HID-1 KO mice show an increase in serum proinsulin levels and glucose intolerance.

Overall, the experiments are well performed and of high quality, particularly the beautiful 3D-EM reconstructions. The conclusions are mostly supported by the data. The reviewers have specific suggestions to improve the manuscript before publications can be recommended.

Essential revisions:

1) While the STX6 data provides an interesting idea about a potential SNARE interaction, much work needs to be done before the HID1-STX6 link can be strengthened. For example, Further evidence for a role for STX6 in homotypic fusion of immature secretory vesicles would be interesting, and evidence that HID1 plays a role would be entirely novel. Similarly, the authors could strengthen their conjecture by demonstrating that an open form of STX6 bypasses a requirement for HID1, or that STX fails to be recruited to immature secretory vesicles in *hid1* mutants or some other functional interaction. Considering the time it will take to do these experiments, the reviewers recommend the authors save the STX6 data for the next paper and eliminate them from the current manuscript.

2) ISGs in *hid1* mutants. The authors argue that in *hid1* KO cells, SGs persist in an immature form and mis-localize throughout the cell. However, the maturation status of the vesicles is monitored only by the presence of pro-insulin cargo and indirectly by size and electron-density in electron micrographs. It is possible that secretory granules in HID1 mutants partially mature. The authors should assess:

Do the secretory granules in *hid1* KO cells retain known immature granule markers including synaptotagmin-IV and VAMP4 (Ahras et al., JCB 2006)?

Do the secretory granules in *hid1* KO cells acquire known markers of maturity, such as synaptobrevin-2 (Walter et al., EMBO 2014)?

3) The Introduction needs to be rewritten with the following points in mind. The specific emphasis on SNAREs in the current Introduction, without the discussion of accessory proteins, is unintentionally misleading. Clarity about HID-1 could be provided by briefly discussing its domain structure and what is known about its ability to associate with membranes. A brief discussion of the granule proteases, ISG acidification, and/or other maturation factors would also be more relevant to the authors findings. Furthermore, HID-1 was also identified in a screen mislocalizing RBF-1 rabphilin, a RAB-27 effector and both of these proteins have important roles in DCV maturation and exocytosis. I think this literature might serve as better introduction to HID-1, and other maturation factors, as opposed to its significance to high-temperature-induced dauer formation and the suppression of HID-1 by the daf-16 transcription factor.

[Editors' note: further revisions were requested prior to acceptance, as described below.]

Thank you for resubmitting your work entitled "HID-1 is required for homotypic fusion of immature secretory granules during maturation" for further consideration at *eLife*. Your revised article has been favorably evaluated by Richard Aldrich as the Senior editor, a Reviewing editor, and three reviewers.

The manuscript has been improved but there are some remaining issues that need to be addressed before acceptance, as outlined below:

All three reviewers felt that the manuscript is significantly improved. There are additional writing suggestions that will for sure make the paper stronger. Please consider these points when you resubmit.

*Reviewer #1:*

The authors have made a number of positive revisions and most of our major concerns were addressed, so we feel the study is ready for publication. Nice paper on a difficult subject – we applaud the authors.

However, there is one small concern that must be addressed: the revision of Figure 5 is confusing because the 3 micron scale bar did not increase in size with the higher magnification. Clearly, there is a minor error here that can be readily corrected.

On a second, also minor note:

As requested by the authors, we do not think the new Figure on syt-4, vamp4, and vamp2 localization should be included in this manuscript. The significance of these experiments seems less clear in light of other reviewers' criticisms of the STX6 data (which has now been removed). A finer resolution of *hid1*'s effects on granule maturation, beyond homotypic fusion, seem beyond the scope of this paper. Additionally, while interesting, these experiments were not conducted as rigorously as other experiments in the paper and are difficult to interpret. For example, these experiments may have been more informative if the analysis was restricted to the perinuclear region (as it was in other experiments) and if they were not maximum projections. It might also be helpful to know more about the deconvolution and see some examples of the raw data to properly evaluate the colocalizaiton data.

*Reviewer #2:*

This is a revised version of a manuscript describing the role of the HID-1 protein in the biogenesis of large dense-core granules (LDCVs) in pancreatic β cells. I will address the rebuttal of the authors point by point.

1) The data on the potential interaction of HID1 with syntaxin6 have been taken out as requested.

2) The authors have performed additional experiments and show that synaptotagminIV, a marker for immature granules, is retained on pro-insulin positive granules in KO cells. In contrast, VAMP4 still remains perinuclear in KO cells. Importantly, VAMP2, a marker for mature vesicles, is not present on proinsulin granules in both WT and KO cells. I like these data and would suggest to include them as supplementary material.

3) The Introduction has been rewritten modestly. Reviewer 1 should decide whether the changes are sufficient.

*Reviewer #3:*

The revised manuscript addresses my concerns from the first review. There are three remaining concerns that warrant textual changes.

1) Homotypic fusion in endocrine cell. The authors argue that HID1 promotes homotypic fusion in β islet endocrine cells. However, the dogma is that homotypic fusion does not occur in endocrine cells, but rather only in neuroendocrine cells. On the other hand, the data against homotypic fusion in endocrine cells are not particularly strong, and that is in part why this manuscript is interesting. The potential conflict with the dogma should be mentioned in the Introduction and expanded upon in the Discussion. Evidence for homotypic fusion is observed in cells that form large dense-core granules such as neuroendocrine cells, including PC12 cells (1,2,3), mammotroph pituitary cells (4), mast cells (5), and in the formation of LDCVs in yeast (6). Evidence for an absence of homotypic fusion in endocrine cells rests on two observations: First, disrupting syntaxin-6 using a dominant negative construct did not exhibit a defect in insulin granule formation in INS-1 β-cells, but rather a delay in pro-insulin processing (7). Second, electron microscopy data in β cells (8, 9) does not reveal an increase in diameter in mature secretory granules. The strengths and weaknesses of these previous data should be mentioned in the Discussion.

2) Granule diameter. The authors claim that previous researchers have noted that mature granules are larger in diameter (and are light density) and cite Furuta 1998, Noske 2008, and Wijesekara 2010 (subsection “HID-1 deficiency increases the number of ISGs”, second paragraph and subsection “HID-1 deficiency blocks homotypic fusion of ISGs”, first paragraph). Furuta 1998 and Wijesekara 2010 both note that immature granules are lighter, but did not report diameters. Noske et al. suggest that old granules are smaller, not larger. In fact, Noske et al. were not studying immature granules, they report a size change in older cells. Orci 1986 also states that maturation involves condensation of the matrix, and reduction in granule diameter. The authors observed the opposite – they see larger mature vesicles suggesting homotypic fusion. The authors may be the first to study granule maturation in any comprehensive way; block in the prohormone convertase or the zinc transporter are unlikely to affect membrane trafficking events.

3) Molecular nature of immature granules. The argument that granules are immature in the *hid1* KO is solid. First, the matrix is lighter. Second, proinsulin is still present in the granules. Third, the presence of immature granule membrane proteins reinforces this conclusion, in particular synaptotagmin 4. Thus, the synaptotagmin 4 data should be included. It is unfortunate that the authors did not include syntaxin 6 staining, to demonstrate that proteins required for homotypic fusion were both still on the pro-insulin granules, which would provide evidence that granule maturation is blocked at a step before homotypic fusion. But that is a quibble; the manuscript should be published. VAMP4 was not mislocalized in the *hid1* KO. This result does not challenge the author's conclusion about a role for HID1 in homotypic fusion. VAMP4 has not been shown to be required for fusion of immature secretory granules, whereas syntaxin-6 is required. The authors conclude that HID1 is required for VAMP4 inclusion in immature secretory granules. Alternatively, VAMP4 may be removed by an intact 'sorting-by-exit' mechanism in the *hid1* KO. These data are an interesting facet of the KO phenotype and should be included. The mature granule marker VAMP2 is not observed on immature granules in *hid1* KO cells. These data are consistent with their model that granules are not mature in *hid1* KO cells. However, the authors did not validate that the VAMP2 antibody was associated with mature insulin granules in the wild type. Thus the negative result is not clearly interpretable and these data unfortunately should not be included.

References

1) Wendler et al. MBC 2001

2) Ahras et al. JCB 2006

3) Tooze SA et al. Trends Cell Biol. 2001

4) Farquar et al. 1981

5) Hammel et al., 2015 Mol. Immunol.

6) Asensio et al. 2013 Dev. Cell

7) Kuliawat et al. MBC 2004

8) Noske et al. 2008

9) Wijesekara et al. 2010

---

## [Author Response]

*[…] Overall, the experiments are well performed and of high quality, particularly the beautiful 3D-EM reconstructions. The conclusions are mostly supported by the data. The reviewers have specific suggestions to improve the manuscript before publications can be recommended.*

*Essential revisions:*

*1) While the STX6 data provides an interesting idea about a potential SNARE interaction, much work needs to be done before the HID1-STX6 link can be strengthened. For example, Further evidence for a role for STX6 in homotypic fusion of immature secretory vesicles would be interesting, and evidence that HID1 plays a role would be entirely novel. Similarly, the authors could strengthen their conjecture by demonstrating that an open form of STX6 bypasses a requirement for HID1, or that STX fails to be recruited to immature secretory vesicles in hid1 mutants or some other functional interaction. Considering the time it will take to do these experiments, the reviewers recommend the authors save the STX6 data for the next paper and eliminate them from the current manuscript.*

We thank the reviewer for this constructive suggestion. In accordance with this suggestion we have eliminated the STX6 data from the current manuscript. We are currently testing the open form of STX6 and how STX6 is recruited to immature dense core vesicles.

*2) ISGs in hid1 mutants. The authors argue that in hid1 KO cells, SGs persist in an immature form and mis-localize throughout the cell. However, the maturation status of the vesicles is monitored only by the presence of pro-insulin cargo and indirectly by size and electron-density in electron micrographs. It is possible that secretory granules in HID1 mutants partially mature. The authors should assess:*

*Do the secretory granules in hid1 KO cells retain known immature granule markers including synaptotagmin-IV and VAMP4 (Ahras et al., JCB 2006)?*

*Do the secretory granules in hid1 KO cells acquire known markers of maturity, such as synaptobrevin-2 (Walter et al., EMBO 2014)?*

As suggested by the reviewer, we investigated whether VAMP4, synaptotagmin IV and VAMP2 were present on proinsulin-positive ISGs employing DeltaVision OMX V3 imaging system followed by deconvolution. As shown in Figure 7, VAMP4 and synaptotagmin IV both localized to the peri-nuclear compartment, similar as the distribution of proinsulin in WT β cells. VAMP4 and synaptotagmin IV partially co-localized with proinsulin puncta to a similar extent (similar Pearson coefficient). In HID-1 KO β cells, proinsulin and synaptotagmin IV staining dispersed throughout the cytosol and co-localized to a similar extent as in WT cells (Figure 7, bottom, B and D). In contrast, the localization of VAMP4 was unchanged and remained in the peri-nuclear region. As a consequence, the co-localization of proinsulin and VAMP4 was significantly reduced in KO cells (Figure 7). Thus, proinsulin-positive secretory granules in HID-1 KO cells seem to retain synaptotagmin IV but not VAMP4, suggesting VAMP4 recruitment or retaining to ISG depends on HID-1. Besides, previous study has shown that VAMP4 antibody did not inhibit homotypic fusion of PC12-derived ISGs, which indicates that VAMP4 is not an essential component for ISG-ISG fusion process (Wendler et al., 2001). As for MSG marker VAMP2, its dispersed localization was not altered by HID-1 KO and there was few co-localization between proinsulin and VAMP2 in WT and KO cells (Figure 7), suggesting VAMP2 is not recruited to proinsulin granules in either WT or KO cells. We are not sure whether to include these data or not and would like to hear the advice from the reviewers.

Author response image 1.Depletion of HID-1 leads to reduced co-localization with VAMP4, but not synaptotagmin-IV and VAMP2.(**A, C, E**) The representative images showing a single β cell stained for Proinsulin (green channel),VAMP4/ synaptotagmin IV / VAMP2 (red channel) and DAPI (blue channel). Each figure shows maximum intensity projection throughout the cell. Top: WT, bottom: KO; Scale bar, 2 µm. (**B, D, F**) Quantitation of pearson's coefficient between green and red channels in A, C and E (n=15, 3 biological replicates, p <0.001, P=0.601, 0.282, respectively, Mann-Whitney *U* test). Scale bar, 2 µm.**DOI:**
http://dx.doi.org/10.7554/eLife.18134.021

*3) The Introduction needs to be rewritten with the following points in mind. The specific emphasis on SNAREs in the current Introduction, without the discussion of accessory proteins, is unintentionally misleading. Clarity about HID-1 could be provided by briefly discussing its domain structure and what is known about its ability to associate with membranes. A brief discussion of the granule proteases, ISG acidification, and/or other maturation factors would also be more relevant to the authors findings. Furthermore, HID-1 was also identified in a screen mislocalizing RBF-1 rabphilin, a RAB-27 effector and both of these proteins have important roles in DCV maturation and exocytosis. I think this literature might serve as better introduction to HID-1, and other maturation factors, as opposed to its significance to high-temperature-induced dauer formation and the suppression of HID-1 by the daf-16 transcription factor.*

We rewrote the Introduction part as suggested.

[Editors' note: further revisions were requested prior to acceptance, as described below.]

*[…] Reviewer #1:*

*The authors have made a number of positive revisions and most of our major concerns were addressed, so we feel the study is ready for publication. Nice paper on a difficult subject – we applaud the authors.*

*However, there is one small concern that must be addressed: the revision of Figure 5 is confusing because the 3 micron scale bar did not increase in size with the higher magnification. Clearly, there is a minor error here that can be readily corrected.*

We are sorry about the mistake. We have increased the length of the scale bar in *Figure 5* accordingly.

*Reviewer #3:*

*The revised manuscript addresses my concerns from the first review. There are three remaining concerns that warrant textual changes.*

*1) Homotypic fusion in endocrine cell. The authors argue that HID1 promotes homotypic fusion in β islet endocrine cells. However, the dogma is that homotypic fusion does not occur in endocrine cells, but rather only in neuroendocrine cells. On the other hand, the data against homotypic fusion in endocrine cells are not particularly strong, and that is in part why this manuscript is interesting. The potential conflict with the dogma should be mentioned in the Introduction and expanded upon in the Discussion. Evidence for homotypic fusion is observed in cells that form large dense-core granules such as neuroendocrine cells, including PC12 cells (1,2,3), mammotroph pituitary cells (4), mast cells (5), and in the formation of LDCVs in yeast (6). Evidence for an absence of homotypic fusion in endocrine cells rests on two observations: First, disrupting syntaxin-6 using a dominant negative construct did not exhibit a defect in insulin granule formation in INS-1 β-cells, but rather a delay in pro-insulin processing (7). Second, electron microscopy data in β cells (8, 9) does not reveal an increase in diameter in mature secretory granules. The strengths and weaknesses of these previous data should be mentioned in the Discussion.*

We thank the reviewer for raising this important dogma and for providing detailed review of the references. As suggested, we have briefly introduced the dogma in the Introduction (first paragraph) and expanded in the Discussion section (subsection “Homotypic fusion in the SG maturation process”, first paragraph).

*2) Granule diameter. The authors claim that previous researchers have noted that mature granules are larger in diameter (and are light density) and cite Furuta 1998, Noske 2008, and Wijesekara 2010 (subsection “HID-1 deficiency increases the number of ISGs”, second paragraph and subsection “HID-1 deficiency blocks homotypic fusion of ISGs”, first paragraph). Furuta 1998 and Wijesekara 2010 both note that immature granules are lighter, but did not report diameters. Noske et al. suggest that old granules are smaller, not larger. In fact, Noske et al. were not studying immature granules, they report a size change in older cells. Orci 1986 also states that maturation involves condensation of the matrix, and reduction in granule diameter. The authors observed the opposite – they see larger mature vesicles suggesting homotypic fusion. The authors may be the first to study granule maturation in any comprehensive way; block in the prohormone convertase or the zinc transporter are unlikely to affect membrane trafficking events.*

We thank the reviewer for bringing up this important point. The reviewer is correct that there is controversy whether mature granules are larger in size or not. The papers we cited (Furuta et al., 1998; Wijesekara et al., 2010) noted the lighter ISG but did not report diameter. So we have changed the sentence to “It has been suggested that ISGs exhibit lighter dense cores than MSGs.” (subsection “HID-1 deficiency increases the number of ISGs”). Tooze et al. (Tooze et al., 1991) have reported the smaller size of ISG in PC12 cells as copied below, so we added this reference in the first paragraph of the subsection “Homotypic fusion in the SG maturation process”.

As the reviewer pointed out, the diameter of granules after maturation may not be always larger. As wrote by Tooze et al. (Tooze et al., 2001), “Maturation of secretory granules also involves a change in size, and MSGs can either become smaller or larger than ISGs. A decrease in size should be proportional to the number of vesicles budded from each ISG, although this has not been confirmed experimentally. MSGs become larger than ISGs through homotypic fusion of ISGs that probably precedes CCV formation”. We have added this possibility in the Discussion section (subsection “Homotypic fusion in the SG maturation process”, first paragraph).

We thank the reviewer for stating that we studied granule size in a comprehensive way. Previous analysis of granule size largely relies on EM of thin sections, which is problematic because planar sectioning at random depths through spherical granules tends to underestimate the diameter of granules and largely increase the variation. As we explained in the text:

“In this study we employed automatic FIB-SEM to visualize large volume (13.7x11.8x3~8 μm^3^) at nanometer resolution and analyze thousands of intact granules from 15 cells. We analyzed not only the diameter, but also the grey level of the cores and the subcellular localization of the granules. This comprehensive analysis enables us to clearly separate ISGs from MSGs and conclude that MSGs have larger size in β cells (Figure 4).”

*3) Molecular nature of immature granules. The argument that granules are immature in the hid1 KO is solid. First, the matrix is lighter. Second, proinsulin is still present in the granules. Third, the presence of immature granule membrane proteins reinforces this conclusion, in particular synaptotagmin 4. Thus, the synaptotagmin 4 data should be included. It is unfortunate that the authors did not include syntaxin 6 staining, to demonstrate that proteins required for homotypic fusion were both still on the pro-insulin granules, which would provide evidence that granule maturation is blocked at a step before homotypic fusion. But that is a quibble; the manuscript should be published. VAMP4 was not mislocalized in the hid1 KO. This result does not challenge the author's conclusion about a role for HID1 in homotypic fusion. VAMP4 has not been shown to be required for fusion of immature secretory granules, whereas syntaxin-6 is required. The authors conclude that HID1 is required for VAMP4 inclusion in immature secretory granules. Alternatively, VAMP4 may be removed by an intact 'sorting-by-exit' mechanism in the hid1 KO. These data are an interesting facet of the KO phenotype and should be included. The mature granule marker VAMP2 is not observed on immature granules in hid1 KO cells. These data are consistent with their model that granules are not mature in hid1 KO cells. However, the authors did not validate that the VAMP2 antibody was associated with mature insulin granules in the wild type. Thus the negative result is not clearly interpretable and these data unfortunately should not be included.*

Reviewer 1 didn’t think the new data should be included, whereas Reviewer 2 suggested including them as supplementary material. Reviewer 3 suggested including Syt-IV and VAMP4, but not VAMP2. We are not sure how we should respond and would like to hear the decision of the editor. For the time being, we included the data of Syt-IV and VAMP4 in the new Figure 4—figure supplement 1, described in the first paragraph of the subsection “HID-1 deficiency increases the number of ISGs”.